# MoCA: Mixture-of-Components Attention for Scalable Compositional 3D Generation

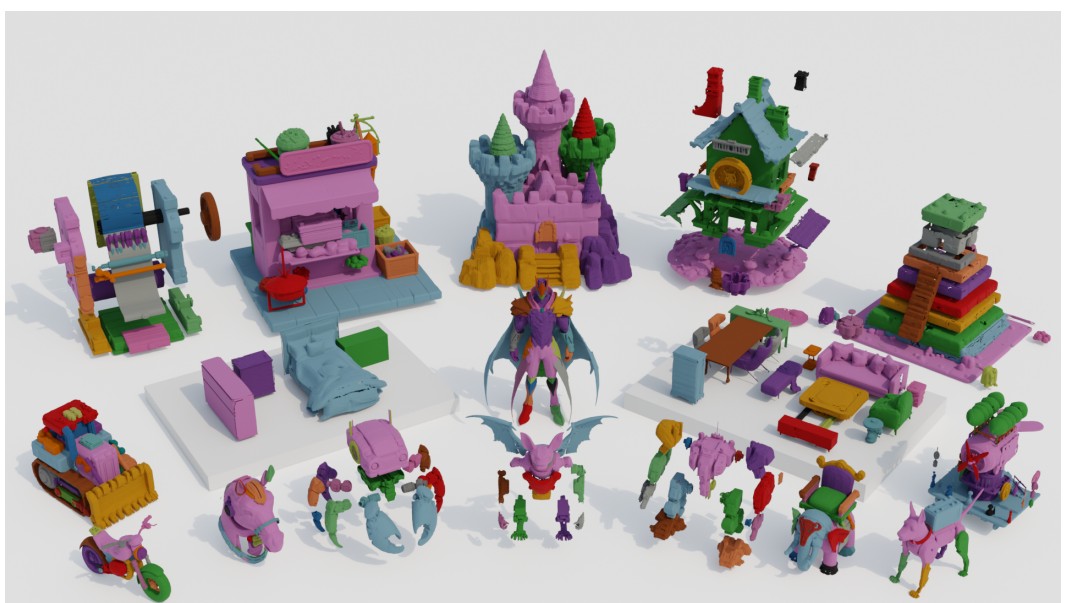

Figure 1: **MoCA** enables scalable compositional 3D generation with up to 32 components per 3D assets. It can generate part-level 3D object with fine-grained composition, and instance-level 3D scenes with complex layout.

## ABSTRACT

Compositionality is critical for 3D object and scene generation, but existing part-aware 3D generation methods suffer from poor scalability due to quadratic global attention costs when increasing the number of components. In this work, we present *MoCA*, a compositional 3D generative model with two key designs: 1) **importance-based component routing** that selects top-k relevant components for sparse global attention, and 2) **unimportant components compression** that preserve contextual priors of unselected components while reducing computational complexity of global attention. With these designs, MoCA enables efficient, fine-grained compositional 3D asset creation with scalable number of components. Extensive experiments show MoCA outperforms baselines on both compositional object and scene generation tasks.

## 1 INTRODUCTION

Compositionality is a a fundamental concept in 3D art design and assets creation pipeline. It describes how complex 3D assets are formed by combining simpler and semantically coherent components, such as objects composed of simple parts and scenes composed of individual objects. These component-aware representations enable the reuse of components, targeted editing, animation, and the modeling of physically plausible interactions, all of which are essential for applications ranging from virtual production and game asset creation to robotics and computer-aided design.

Recently, 3D diffusion transformer (DiT) models (Zhang et al., 2024; Li et al., 2025a; Zhao et al., 2025; Xiang et al., 2024) have dramatically improved the generation quality of an individual object. However, these models commonly treat an object or scene as a monolithic entity, which limits controllability over the generated content and makes many downstream tasks (*e.g.*, component-level editing, materials customization for each component) challenging or even unachievable. Recent works (Huang et al., 2025; Lin et al., 2025) explore compositional 3D generation by extending pre-trained 3D shape generators (Li et al., 2025a; Hunyuan3D et al., 2025; Xiang et al., 2024) to jointly generate multiple components (either parts or instances).

While these works demonstrate that structured latent spaces enable the generation of a few semantically coherent parts and multi-object scenes, they suffer from two critical limitations when scaling up the number of components: 1) *Salability of global attention across components.* To model cross-component dependencies, previous models for multi-component generation often leverage global attention modules over all component tokens by concatenating their token sets. In this naive design, $N$ components (each represented by $L$ latent tokens) result in a global attention over $N \times L$ tokens with quadratic computational cost $O(N^2 L^2)$. As $N$ grows (*e.g.* a complex scene, fine-grained decompositions of an object), this cost quickly becomes computationally prohibitive. **2)** *Uniform attention wastes capacity.* Not all components exhibit strong interactions with one another. For example, modeling a character's hand typically only requires detailed information from the wrist and forearm, while modeling the position of a chair primarily correlates with the nearby table. Blindly enabling every token to attend to all others allocates computational resources and memory to numerous low-value interactions, leading to inefficiency and constraining model scale.

To address these problems, we propose **MoCA**, a native compositional 3D generative model equipped with a novel **M**ixture-**o**f-**C**omponents **A**ttention mechanism, designed for efficient, scalable, and accurate compositional modeling of 3D objects and scenes. MoCA is built upon two key designs:

- **Importance-based component routing.** Inspired by the practices of MoE (Mixture-of-Experts) models (Fei et al., 2024; Dai et al., 2024; Jiang et al., 2024), we introduce a lightweight router module within the global block. For a given component, the router estimates the importance of other components relative to it, and then selects the top-$k$ important components for sparse global attention. This design is based on the hypothesis that *a given component only requires detailed information from a small subset of other components*.

- **Compression of distant components.** Unlike previous methods, for the components not selected by the router, we also compress them in compact tokens for global attention rather than discarding them. This preserves coarse-grained context (e.g., spatial priors, presence/absence cues) while dramatically reducing the number of key/value tokens in attention computations.

The combination of the these two designs yields a context length of $L_{global} = L + kL + (N-k-1)\frac{L}{\sigma}$ in our global attention layers, where $\sigma$ is the compression ratio of unimportant components. With typical settings where $k \ll N$ and $\sigma \gg 1$, the context length is much smaller than the naive global attention used in prior works (Huang et al., 2025; Lin et al., 2025), enabling efficient yet powerful compositional 3D generation.

We evaluate MoCA across both object-level and scene-level 3D generation tasks. For object-level generation, MoCA generates a 3D object from a single image with automatically decomposed 3D parts. For scene-level generation, it produces instance-composed 3D scenes conditioned on scene images, using per-instance masks as auxiliary conditions. Extensive qualitative and quantitative experiments demonstrate that MoCA outperforms prior works by a clear margin, with particularly notable improvements in fine-grained component generation.

## 2 RELATED WORKS

**3D Latent Diffusion Models.** Recent approaches that extend latent diffusion models (LDMs) to native 3D shape generation can be broadly categorized into two paradigms: vecset-based methods and sparse voxel-based methods. Vecset-based methods leverage a compact latent representation introduced by 3DShape2VecSet (Zhang et al., 2023). Subsequent studies (Zhang et al., 2024; Li et al., 2024; Wu et al., 2024; Li et al., 2025a; Zhao et al., 2025) have advanced this paradigm to generate 3D shapes with high-resolution details, demonstrating its scalability and representational capacity. In contrast, Trellis (Xiang et al., 2024) proposes a structured latent space grounded in sparse voxels.

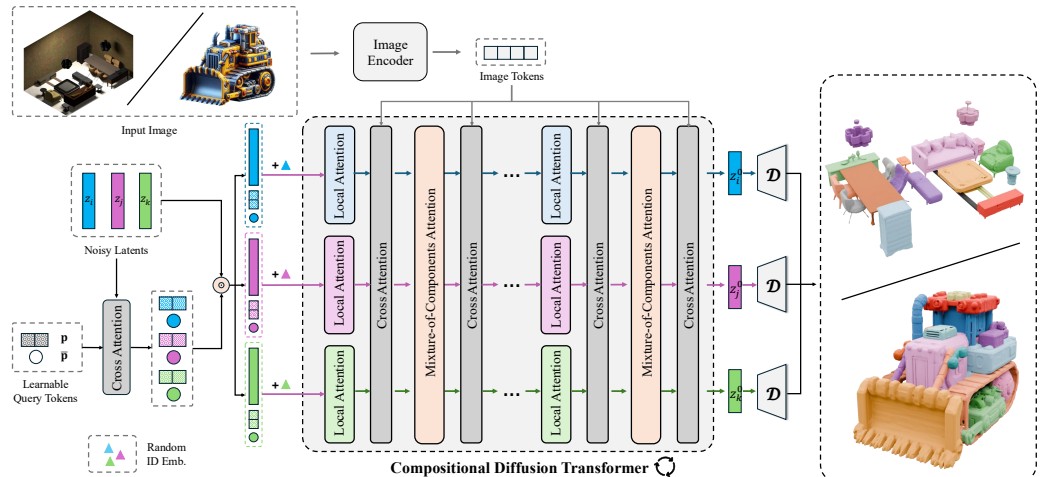

Figure 2: **Overview of MoCA**. Our DiT model starts with packing each component's latents using several learnable queries through a cross-attention layer. Random ID embeddings are applied to distinguish different components. Then, each component's full latents and compressed version are fed into our DiT model, which is comprised with interleaved local attention blocks and our proposed Mixture-of-Components Attention blocks. Finally, the clean latents of all components are separately decoded to the global space by a frozen shape decoder to form the final 3D asset.

Follow-up work (Ye et al., 2025; He et al., 2025; Wu et al., 2025; Li et al., 2025b) shows that such sparse voxel-based latent spaces excel at capturing fine-grained geometric details. Both vecset and sparse voxel-based representation produce implicit occupancy or SDF fields rather than explicit 3D meshes, hence they require an iso-surface extraction step, such as Marching Cubes (Lorensen & Cline, 1998), to obtain triangular meshes.

**Part-Aware 3D Object Generation.** While 3D generative models have demonstrated remarkable capabilities in producing high-quality 3D objects (Hong et al., 2024; Wang et al., 2023; Tang et al., 2024; Wang et al., 2024b), a significant challenge lies in generating part-aware 3D objects with separate components, which are necessary downstream tasks like 3D editing. Early methods (Liu et al., 2024b; Chen et al., 2024a) leveraged pretrained 2D priors (Kirillov et al., 2023; Liu et al., 2023) to generate multi-view images with part segmentation and then reconstruct them to 3D. Another line of work (Yang et al., 2025) decomposes the generation process into 3D segmentation followed by 3D part completion, but the generated parts are often difficult to assemble seamlessly. More recent works (Lin et al., 2025; Tang et al., 2025; Chen et al., 2025; Yan et al., 2025) train 3D shape generation models on preprocessed 3D part datasets, achieving higher-quality geometry in part-aware generation. They typically denoise token sequences of $N$ parts concurrently, using global attention to model their inter-part composability. However, as $N$ grows, the quadratic increase in computational cost makes these frameworks difficult to scale. To address this limitation, we introduce a scalable solution that efficiently generates objects with more parts while significantly reducing computational overhead.

**Compositional 3D Scene Generation.** 3D scene generation plays a significant role in gaming and simulation, but is challenging because it requires modeling the geometry of individual objects and the complex spatial relationships between them (Chu et al., 2023; Lin & Mu, 2024; Han et al., 2024; Tencent, 2025; Yu et al., 2024). Among these, multi-instance 3D scene generation (Ardelean et al., 2024; Yao et al., 2025; Ni et al., 2025; Meng et al., 2025; Zhou et al., 2024; Lin et al., 2024) has become a promising direction, focusing on the creation of multiple independent and well-arranged objects. Some methods (Chen et al., 2024b; Ardelean et al., 2024; Yao et al., 2025) generate each object in a scene sequentially and then optimize their layout to compose the scene. However, this multi-stage approach is lengthy and inefficient. Recent methods (Huang et al., 2025), adopted an end-to-end paradigm, which generates all 3D objects simultaneously and employs global attention in diffusion transformers to model spatial relationships. However, these approaches are constrained in the number of instances they can effectively handle, typically fewer than 20. This limitation motivates our work, which focuses on enhancing the scalability of such pipelines with an efficient attention mechanism.

## 3 MoCA

### 3.1 Preliminary: Vecset Diffusion Models for 3D Shape Generation

Vecset diffusion models (Zhang et al., 2024; Zhao et al., 2025; Li et al., 2025a) are a class of latent diffusion models trained to generate a set of unordered vectors (vecset) which implicitly encapsulate a 3D shape. The vecset VAE (Zhang et al., 2023) consists of several key steps:

- Surface Sampling: Dense points $P_d \in \mathbb{R}^{N_d \times 3}$ are sampled on the shape surface uniformly or from sharp edge region. Then, sparse points $P_s \in \mathbb{R}^{N_s \times 3}$ are obtained by downsampling $P_d$ using farthest point sampling (FPS).

- Encoding: The sparse points $P_s$ firstly aggregates features from $P_d$ through a cross-attention layer, then go through a sequence of self-attention layers to obtain the latents $\mathbf{z} \in \mathbb{R}^{N_s \times D}$:

$$\mathbf{z} = \texttt{SelfAttn}\left(\texttt{CrossAttn}\left(P_s, P_d, P_d\right)\right). \tag{1}$$

- Decoding: The decoder is of symmetric architecture as encoder. The latents $z$ firstly go through several self-attention layers to form a implicit field, from where the occupancy or SDF values at any spatial point $p \in \mathbb{R}^3$ can be queried through a cross-attention layer:

$$h = \texttt{SelfAttn}\left(\mathbf{z}\right), \hat{o} = \texttt{CrossAttn}\left(p, h, h\right). \tag{2}$$

- Surface Extraction: The explicit surface of shape can be extracted using classic iso-surface extraction methods, e.g., Marching Cubes (Lorensen & Cline, 1998).

### 3.2 Model Architecture

As illustrated in Figure 2, the core of MoCA 's architecture is a compositional diffusion transformer (DiT) (Peebles & Xie, 2023) with interleaved local and global attention blocks. Specifically, conditioned on an input image $\mathcal{I}$ of an object or scene, our DiT model jointly generates clean vecset latents $\mathcal{Z} = \{\mathbf{z}_i\}_{i=1}^N \in \mathbb{R}^{N \times L \times D}$. These latents are then decoded into individual 3D components using a frozen vecset decoder: $\mathbf{c}_i = \mathcal{D}(\mathbf{z}_i)$. By integrating all components, we can obtain the target 3D asset.

To enable the computation of our proposed Mixture-of-Components (MoC) attention within global blocks, we append several learnable tokens to the noisy latents of each component. These tokens act as queries that compress the vecset features through a cross-attention layer, as described in Section 3.2.1. Within each local block (Section 3.2.2), both the vecset tokens and their compressed representations are updated. In the global blocks, MoC attention facilitates inter-component communication: each component attends to the full-token features of relatively important components and the compressed features of less important ones (Section 3.2.3)

#### 3.2.1 Learnable Tokens and Component Compression

**Learnable Query Tokens.** Unlike images or videos, which can be efficiently compressed using convolutions or pooling, the unstructured and unordered nature of vecset latents makes them suitable only for compression via attention-based query operations. To this end, we append $N_p = \frac{L}{\sigma}$ learnable tokens $\mathbf{p} \in \mathbb{R}^{N_p \times D}$ to the noisy latents of each component. To further support the computation of inter-component importance (introduced in Section 3.2.3), we also append an additional learnable token $\bar{\mathbf{p}} \in \mathbb{R}^{1 \times D}$, which aggregates the features of each component into a single token. This token serves as an anchor for its corresponding component.

**Cross-Attention Packing.** Before feeding tokens into the DiT model, we employ a cross-attention layer to obtain compressed representations for each component:

$$\mathbf{p}_i = \texttt{CrossAttn}(\mathbf{p}, \mathbf{z}_i, \mathbf{z}_i), \quad \bar{\mathbf{p}}_i = \texttt{CrossAttn}(\bar{\mathbf{p}}, \mathbf{z}_i, \mathbf{z}_i), \tag{3}$$

where $\mathbf{p}_i$ and $\bar{\mathbf{p}}_i$ denote the compressed tokens and anchor token of the $i$-th component, respectively. For clarity, we omit the diffusion timestep $t$ in the notation.

**All Input Tokens.** The final input tokens for the $i$-th component are formed by concatenating its noisy vecset tokens $\mathbf{z}_i$, the compressed tokens $\mathbf{p}_i$, and its anchor token $\bar{\mathbf{p}}_i$: $\mathbf{x}_i = \texttt{Concat}(\mathbf{z}_i; \mathbf{p}_i; \bar{\mathbf{p}}_i)$. To distinguish different components, we further add random ID embeddings to all input tokens.

### 3.2.2 LOCAL BLOCK: LOCAL FEATURE UPDATING

In each local block, all input tokens update their features within a local scope. To preserve the original shape modeling capacity of the pretrained prior, the vecset tokens $z_i$ are restricted to self-attention only, remaining blind to the newly appended tokens. For the compressed tokens $p_i$, we allow attention over both the vecset tokens and other compressed tokens. This design enables them to refine their features by aggregating information from the vecset tokens while maintaining coherence with the other compressed tokens. Finally, the single anchor token $\bar{p}_i$, as the highest-level abstraction of a component's features, is granted access to all tokens in $x_i$. This feature updating strategy results in a partially blocked causal attention mask.

### 3.2.3 GLOBAL BLOCK: MIXTURE-OF-COMPONENTS ATTENTION

During the global blocks, we perform Mixture-of-Components (MoC) attention to enable inter-component communication. Here, we focus on how a specific component $c_i$ attends to other components and computes global attention. The forward stream of $c_i$ is illustrated in Figure 3. This procedure is permutation-invariant across all components.

Inspired by Mixture-of-Experts (MoE) models (Fei et al., 2024; Dai et al., 2024; Jiang et al., 2024), where router modules determine which top-$k$ experts are selected for each token, we introduce a router module to estimate the relative importance of all other components with respect to $c_i$. Based on these importance scores, the router decides whether each component should be attended through its full-token representation or through its compressed version.

**Importance Computation Through Component-Level Attention.** The importance of a component quantifies how much attention component $c_i$ should allocate to it. Naturally, this can be computed in a component-level attention-like manner.

Specifically, to compute the importance of component $c_j$ with respect to $c_i$, we project the anchor token of $c_i$ into a query vector $\bar{Q}_i = \text{Query}(\bar{p}_i)$, and the anchor token of $c_j$ into a key vector $\bar{K}_j = \text{Key}(\bar{p}_j)$, using two separate linear projection layers. The importance score is then given by:

$$o_{i,j} = \text{Sigmoid}\left(\frac{\bar{Q}_i \bar{K}_j^T}{\sqrt{d_{\bar{K}}}}\right), \quad (4)$$

where $d_{\bar{K}}$ denotes the dimensionality of the projected key vectors. Instead of the conventional $\text{Softmax}(\cdot)$, we adopt $\text{Sigmoid}(\cdot)$ as the activation function, which helps avoid indistinguishable logits when $N$ is large, following a similar design choice to DeepSeek-V3 (Liu et al., 2024a).

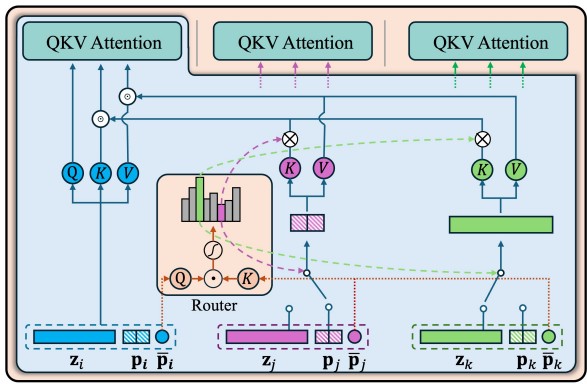

Figure 3: **Illustration of Mixture-of-Components Attention**. The calculation stream for component $c_i$ is highlighted. This procedure is permutation-invariant across all components.

**Routing Based on Importance.** After computing all importance scores $\{o_{i,j}\}_{j \neq i}$, we select the top-$k$ most important components relative to $c_i$. Let $r_{i,j}$ denote the tokens of component $c_j$ attended by $c_i$. Formally,

$$r_{i,j} = \begin{cases} z_j, & \text{if } o_{i,j} \in \text{TopK}\left(\{o_{i,j}\}_{j \neq i}\right), \\ p_j, & \text{otherwise.} \end{cases} \quad (5)$$

In other words, $c_i$ attends to the full vecset tokens $z_j$ for the top-$k$ important components, while attending only to the compressed tokens $p_j$ for the remaining ones.

**Mixture-of-Components Attention.** The context tokens that $c_i$ attends to in the global block are formed by concatenating its own vecset tokens $z_i$ with the routed tokens from all other components. To further modulate the contributions of different components, we apply the predicted importance scores as gating factors to the key vectors of their corresponding routed tokens during attention computation. Formally, the query, key and value for $c_i$ in this MoC attention are defined as:

Figure 4: **Qualitative comparison for part-composed object generation**. PartPacker can not control the number of generated parts and tends to generate coarse-grained decomposition. PartCrafter suffers from poor surface quality and large-area floaters on complex composition. We run PartCrafter with the same part number configuration as ours.

$$\mathcal{Q}_i = \texttt{Query}(\mathbf{x}_i), \tag{6}$$

$$\mathcal{K}_i = \texttt{Concat}\left(\texttt{Key}(\mathbf{z}_i); o_{i,1} \cdot \texttt{Key}(\mathbf{r}_{i,1}); \ldots; o_{i,N} \cdot \texttt{Key}(\mathbf{r}_{i,N})\right), \tag{7}$$

$$\mathcal{V}_i = \texttt{Concat}\left(\texttt{Value}(\mathbf{z}_i); \texttt{Value}(\mathbf{r}_{i,1}); \ldots; \texttt{Value}(\mathbf{r}_{i,N})\right). \tag{8}$$

By multiplying the predicted importance scores with the corresponding key vectors, the attention allocated to each component is reweighted at the component level, allowing the router layers to be trained end-to-end via backpropagation.

**Multi-Head Routing.** To capture diverse inter-component dependencies and enhance the model's representational capacity, the routing procedure is applied independently across different heads of the multi-head attention.

| Method | PartObjaverse (Yang et al., 2024) | | | | ABO (Collins et al., 2022) | | | |
|---|---|---|---|---|---|---|---|---|
| | Self-IoU↓ | CD↓ | Fscore-0.1↑ | Fscore-0.05↑ | Self-IoU↓ | CD↓ | Fscore-0.1↑ | Fscore-0.05↑ |
| HoloPart | 0.0142 | 0.1145 | 0.8340 | 0.6671 | 0.0139 | 0.1168 | 0.8523 | 0.6772 |
| PartPacker | **0.0120** | 0.1105 | 0.8484 | 0.6510 | 0.0119 | 0.1094 | 0.8646 | 0.6801 |
| PartCrafter* | 0.0224 | 0.1195 | 0.8169 | 0.6236 | 0.0136 | 0.1124 | 0.8515 | 0.6759 |
| Ours | 0.0125 | **0.1010** | **0.8708** | **0.6882** | **0.0116** | **0.1027** | **0.8755** | **0.6871** |

Table 1: **Quantitative results for part-composed object generation**. We asses all the compared methods on PartObjaverse-Tiny dataset.

## 3.3 TRAINING

### 3.3.1 LOAD BALANCE CONSIDERATION

The load imbalance issue in MoE researches refers to the model always selects only a few experts, preventing other experts from sufficient training (Shazeer et al., 2017; Dai et al., 2024). In our case, the risk of load imbalance also exists when routing the components by always deterministically pick the top-$k$ important ones. It would cause two notable defects. First, one component will overly rely on several specific components but overlook others, thus preventing the model from learning diverse inter-component dependencies and constraining the model's representation capacity. Second, if some unimportant components were unexpectedly assigned as important in the early stage of training, the model would refuel those unimportant components and push itself towards suboptimal.

A common practice in the field of MoE to address the imbalance issue is adding a load-balancing auxiliary loss (Shazeer et al., 2017; Lepikhin et al., 2020; Fedus et al., 2022) that encourages the router to use experts more evenly. However, the strength of this auxiliary loss is hard to be determined and introduce undesired gradients (Wang et al., 2024a). Here, we propose a simple-but-effect auxiliary-loss-free solution to ensure load balance of component routing. In particular, we choose the indices of the $k$ full-information components for $\mathbf{c}_i$ by sampling from a probabilistic distribution constructed by normalizing the importance logits $\{o_{i,j}\}_{j \neq i}$. This stochastic routing encourages exploration and mitigates collapse onto a few components, and the predicted importance scores are accordingly corrected in the training process. At test time we revert to deterministic routing for stable inference.

### 3.3.2 TRAINING OBJECTIVE

MoCA is trained with rectified flow matching objective (Esser et al., 2024; Liu et al., 2022). Denoting the clean vecset latents of all components as $\mathcal{Z}_0 = \{\mathbf{z}_i^0\}_{i=1}^N \in \mathbb{R}^{N \times L \times D}$, for each training step, we perturb them with Gaussian noise $\epsilon \sim \mathcal{N}(0, \mathbf{I})$ towards a shared noise level $t \sim \mathcal{U}(0, 1)$ along a linear trajectory: $\mathcal{Z}_t = (1 - t)\mathcal{Z}_0 + t\epsilon$. With $\mathcal{Z}_t$ as input, the model is trained to predict the velocity term $\epsilon - \mathcal{Z}_0$ conditioning on the noise level $t$ and conditioning image $\mathcal{I}$ by minimizing the below objective:

$$\mathcal{L} = \mathbb{E}_{\mathcal{Z}, \epsilon, t} \left[ \|(\epsilon - \mathcal{Z}_0) - \mathbf{v}_\theta(\mathcal{Z}_t, t, \mathcal{I})\|^2 \right], \tag{9}$$

where $\mathbf{v}_\theta$ is the predicted velocity. During training, we randomly drop the condition $\mathcal{I}$ with $10\%$ chance for classifier-free guidance (Ho & Salimans, 2021).

## 4 EXPERIMENTS

### 4.1 IMPLEMENTATION DETAILS

We alternately use self-attention and MoC attention across different layers in our DiT model. During global blocks, $25\%$ components are selected as important for per sample, while the remaining unimportant components are compressed with ratio $\sigma = 8$. The size of random ID embedding codebook is set to 50. For the details of dataset curation and training, please refer to Appendix A.2.

### 4.2 PART-COMPOSED 3D OBJECT GENERATION

**Evaluation Protocol.** We evaluate our model on PartObjaverse-Tiny (Yang et al., 2024) which comprises 200 objects from various categories, and randomly sampled 100 objects from ABO dataset

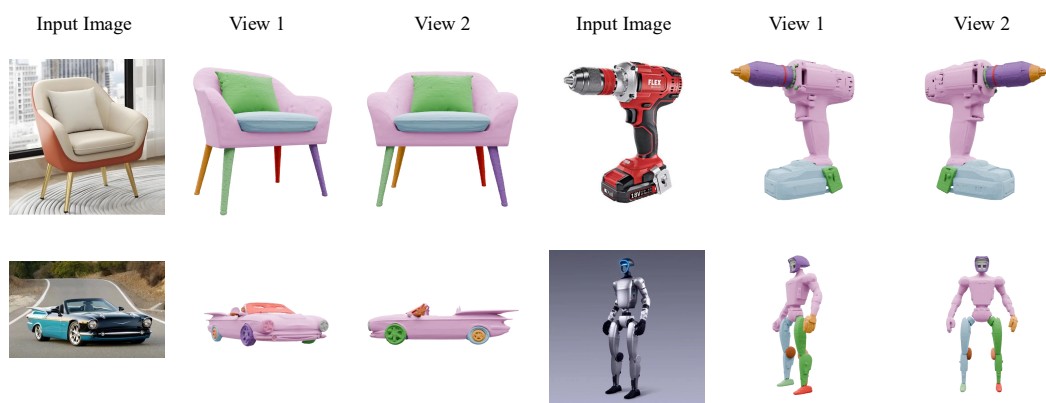

| Input Image | View 1 | View 2 | Input Image | View 1 | View 2 |

Figure 5: **Qualitative results on real-world images.**

(Collins et al., 2022). While we do not have the correspondence between generated and ground truth parts, we focus on assess the overall geometric quality of generated object. In particular, we compute Chamfer Distance (CD) and F-score on fused surface points of all parts. The F-score is computed at two different thresholds [0.1, 0.05] to capture both coarse- and fine-level geometric alignment. In addition, we utilize self-IoU (Intersection over Union) to assess the intersection between parts.

**Results.**  As shown in Figure 4, our model surpasses two compared native part-level object generation methods, PartPacker (Tang et al., 2025) and PartCrafter (Lin et al., 2025), with finer granularity and cleaner decomposition. As for the quantitative evaluation, we additionally compare to HoloPart (Yang et al., 2025). We utilize TripoSG (Li et al., 2025a) to generate the shape firstly, then obtain the 3D mask through SAMPart3D (Yang et al., 2024) for HoloPart. Especially, since PartCrafter is trained exclusively on front-view images, we evaluate it with all front-view renderings to preserve its performance, whereas all other methods all conditioned on random-view renderings. The results in Table 1 suggest that MoCA achieves superior inter-part independence and overall fidelity compared to all baselines.

**Generalization to Real Images**  We further evaluate our model conditioning on real images. Several visual results are shown in Figure 5, where we successfully generate clean part-level decomposition for real-world objects covering different categories. It demonstrates the potentials of our model in generating real-world 3D objects in part level.

### 4.3 INSTANCE-COMPOSED 3D SCENE GENERATION

**Evaluation Protocol.**  We utilize the test split of 3D-FRONT (Fu et al., 2021) processed by MIDI (Huang et al., 2025) for evaluation. Besides of self-IoU, we compute both scene-level and instance-level CD and F-score (with threshold 0.1) against the ground truth meshes.

| Method | Self-IoU↓ | CD-Scene↓ | Fscore-Scene↑ | CD-Obj↓ | Fscore-Obj↑ |
|---|---|---|---|---|---|
| MIDI | 0.0011 | 0.1548 | 0.8238 | 0.1442 | 0.7996 |
| PartCrafter | 0.0036 | 0.1366 | 0.8379 | - | - |
| Ours | **0.0006** | **0.1201** | **0.8637** | **0.1163** | **0.8380** |

Table 2: **Quantitative results for instance-composed scene generation**. We asses all the compared methods on 3D-FRONT (Fu et al., 2021) dataset.

**Results.**  We compare our method with MIDI (Huang et al., 2025) and PartCrafter (Lin et al., 2025). The results in Table 2 demonstrate that our method outperforms both baselines at the scene and instance levels. As illustrated in Figure 7, our method consistently produces higher-quality geometry, with instances positioned more accurately according to the input image. In contrast, PartCrafter, which does not take instance masks as input, can generate duplicate or incorrectly placed instances. Furthermore, we provide qualitative examples of complex scene generation (with >16 instances) in Figure 6, highlighting the scalability of our model, whereas the baseline methods are limited to fewer than 8 instances per scene.

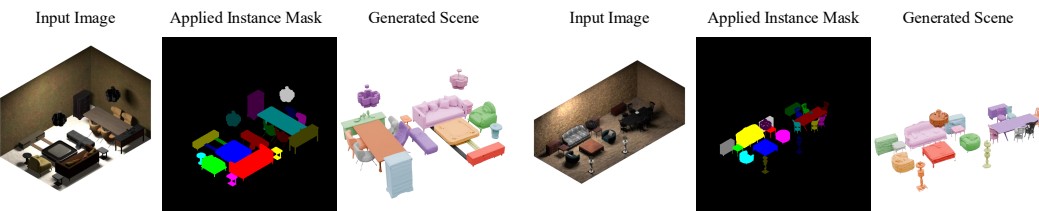

Figure 6: **Complex scene (>16 instances) generation.** MoCA has the expertise to generate complex scene from a single image, which capacity previous scene generation methods do not possess.

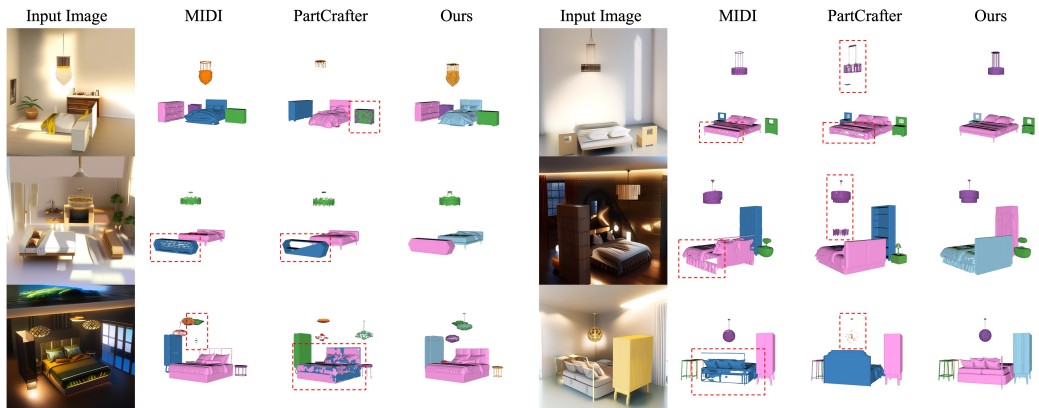

Figure 7: **Qualitative comparison on the generation of simple scenes (<8 instances).** The baseline methods suffer from broken surface frequently. Without the aids of instance masks, PartCrafter has the risk of confusing the identities of instances and generating overlapping objects.

### 4.4 ABLATIONS

We conduct ablations on Trellis-500K (Xiang et al., 2024) dataset with up to 16 parts for each object and the vecset length as 512. Then we evaluate the performance of different design choices on PartObjaverse-Tiny (Yang et al., 2024) dataset. We present the quantitative results in Table 3. For the qualitative cases please refer to Figure 8.

**Generation with Different Granularity.** By specifying varied number of components, we can generate 3D assets with decomposition under different granularity, as shown in Figure 9.

**Necessity of routing to important components.** To evaluate the benefits of including full features of predicted-important components during global attention, we remove the routing process in our model and make the global attention computed on only compressed component features. As shown in Table 3O, the generation quality degrades largely without the interaction with important components' full features.

**Necessity of compressed distant components.** As shown in Table 3A, performance drops significantly without the coarse information from less important components. This indicates that each component benefits from global spatial context provided by the coarse layout of all other components, in addition to the full but local features of the most important ones.

**What if multiply the router logits to value?** In our method, the importance scores are multiplied with the key vectors of the corresponding components, effectively serving as pre-weighting before the softmax operation during attention. We also experimented with multiplying the logits directly with the value vectors, which resulted in a substantial performance drop, as shown in Table 3B. This degradation is likely due to the post-softmax weighting causing the sum of attention weights to deviate from 1, introducing numerical instability in the QKV attention computation.

| Configuration | Routing to Important Components | Compressed Distant Components | Router Logits Multiplied to | Router Logits Activation | Load Balance | Multi-Head Routing | CD ↓ | Fscore-0.1 ↑ | Fscore-0.05 ↑ |
|---|---|---|---|---|---|---|---|---|---|
| O | ✗ | ✓ | Key | Sigmoid | ✓ | ✓ | 0.1969 | 0.6604 | 0.4750 |
| A | ✓ | ✗ | Key | Sigmoid | ✓ | ✓ | 0.1523 | 0.7494 | 0.5465 |
| B | ✓ | ✓ | Value | Sigmoid | ✓ | ✓ | 0.1519 | 0.7612 | 0.5531 |
| C | ✓ | ✓ | Key | Softmax | ✓ | ✓ | 0.1451 | 0.7638 | 0.5539 |
| D | ✓ | ✓ | Key | Sigmoid | ✗ | ✓ | 0.1361 | 0.7947 | 0.5975 |
| E | ✓ | ✓ | Key | Sigmoid | ✓ | ✗ | 0.1212 | 0.8215 | 0.6178 |
| F (Full Model) | ✓ | ✓ | Key | Sigmoid | ✓ | ✓ | **0.1180** | **0.8259** | **0.6233** |

Table 3: **Ablation Study for MoCA**. We conduct training for all settings on Trellis-500K (Xiang et al., 2024) dataset, and evaluate on PartObjaverse (Yang et al., 2024).

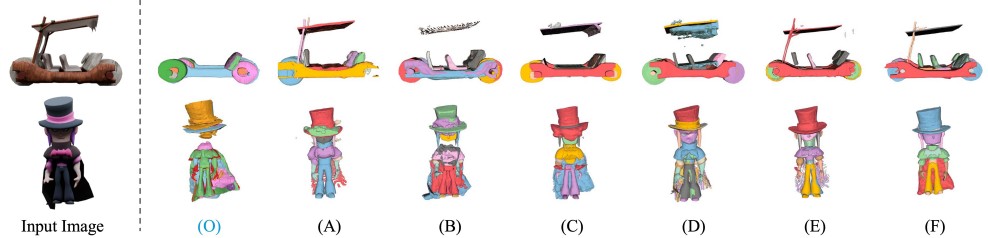

Input Image    (O)    (A)    (B)    (C)    (D)    (E)    (F)

Figure 8: **Qualitative results of ablations**. The full model generates compositional 3D assets with best geometry quality and component decomposition.

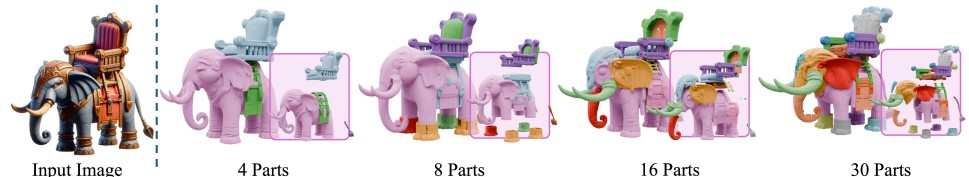

Input Image    4 Parts    8 Parts    16 Parts    30 Parts

Figure 9: **Qualitative results of varied decomposition granularity**.

**Sigmoid vs. Softmax as router logits activation.** As discussed before, using $\texttt{Softmax}(\cdot)$ as activation of router logits leads to very small gate values for the key vectors, which causes vanishing influence from other components, and largely degenerates the generation performance (Table 3C).

**Benefits of load balance consideration.** Ensuring load balance prevents components from relying excessively on a few specific components and encourages more diverse routing during training. The performance improvement shown in Table 3D highlights the benefits of addressing load imbalance.

**Effects of multi-head routing.** Analogous to multi-head attention, scaling our routing mechanism along the head dimension allows each token-level attention head to learn component-level dependencies independently. This strategy enhances the model's representational capacity, as reflected by the performance drop shown in Table 3E when it is removed.

## 5 CONCLUSION

In this work, we propose MoCA, a novel native compositional 3D generator featuring a tailored Mixture-of-Components (MoC) Attention mechanism for scalable training. MoCA is applicable to both part-composed object generation and instance-composed scene generation, consistently outperforming baseline methods across both tasks. Leveraging the efficiency of MoC attention, our model can be trained with up to 32 components per training sample—twice the capacity of previous methods—demonstrating a particular strength in modeling complex 3D assets and establishing a new frontier in native compositional 3D generation.

**Limitations and Future Works** Following previous methods, we make the vecset VAE frozen. It could be a bottleneck when there are large number of components within an asset, since the volume of each component could be very small in the global space. We schedule to further finetune the VAE using component-level data for better reconstruction performance in the future works.

ETHICS STATEMENT

Our work proposes a scalable compositional 3D generation framework aiming to address the computation bottleneck that exists in the global attention blocks of previous methods, by developing a novel efficient and more interpretable Mixture-of-Components mechanism for global attention calculation. We do not anticipate any direct negative societal consequences from this research. However, as with many machine learning methods, potential downstream applications may raise ethical concerns. We encourage careful consideration and responsible use of our methods in applied settings. All data utilized for training are curated from a blend of public and professionally sources, followed by a rule-based filtering. The resulting dataset does not involve human subjects, personally identifiable information, or sensitive data.

REPRODUCIBILITY STATEMENT

The full model is scheduled to be released for community reproduction. For the stage that our model has not been released, we also have provided the elaborate experimental configurations including the hyperparameters of training, the detailed model architecture, the computation resources utilized, in the Appendix. With these materials, the independent researchers should be able to achieve a comparable performance according to using the open-source data.

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

# A   APPENDIX

## A.1   LLM USAGE STATEMENT

The authors declare that ChatGPT was used exclusively for grammar checking and stylistic refinement of the manuscript text. All scientific content—including the conception of research ideas, experimental design, data collection, data analysis, and formulation of conclusions—was entirely the original work of the authors. The language model was not utilized for generating scientific hypotheses, conducting experiments, interpreting findings, or drawing conclusions.

## A.2   IMPLEMENTATION DETAILS

**Dataset Curation.**   For part-composed object generation, we train our model on the dataset blended from public and professionally sources, leading to nearly 2 million watertight meshes with part-level annotations. We utilize the objects with up to 32 parts and render image from random views for training. As for object-composed scenes generation, we adapt the object model to 3D-FRONT (Fu et al., 2021) dataset for up to 32 instances in each scene. When many instances exist in a room, it is hard to capture all the objects in a simple perspective view. Therefore, for each scene, we append randomly textured floor and wall to it and render four isometric view images that clearly display all objects in the room. We mix the processed 3D-Front data in MIDI (Huang et al., 2025) with our newly rendered isometric image to train our scene model.

**Training.**   To train our object model, we firstly pretrain it on our object dataset by setting the length of vecset latents as 512 with a learning rate of 1e-4 for 40K steps, then we finetune the model with vecset length as 1024 for another 10K steps, under learning rate 4e-5. The training is carried on 32 H20 GPUS with 32 parts in each, leading to a total batch size of 1024. Then we adapt the object model to scene generation task by finetuning it on the scene dataset for another 10K steps with learning rate 4e-5 on 8 H20 GPUs. During scene task training, we use both the scene image and instance masks as condition, which are concatenated along channel dimension. Therefore, following (Huang et al., 2025), we modify the input channel of image encoder to 7, and finetune it using LoRA (Hu et al., 2022) along with DiT. During all training procedure, we randomly drop the image condition with probability 0.1 for CFG (Ho & Salimans, 2021). We utilize AdamW as our optimizer, and the training precision is `BFloat16`.

## A.3   RUNTIME ANALYSIS

Table 4: **Runtime analysis of MoCA and PartCrafter (Lin et al., 2025) under different number of parts (unit: ms)**. We report the breakdown runtime of local attention, routing procedure (MoCA only), and global attention respectively, along with the total runtime of a single forward of the model.

| Number of Parts | Method | Local Attention | Routing | Global Attention | Total |
|---|---|---|---|---|---|
| N=4 | MoCA | 6.9 | 1.7 | 7.5 | 180 |
| | PartCrafter | 6.0 | / | 7.8 | 146 |
| N=8 | MoCA | 12 | 4.2 | 14 | 327 |
| | PartCrafter | 12 | / | 18 | 320 |
| N=16 | MoCA | 25 | 12 | 31 | 689 |
| | PartCrafter | 25 | / | 49 | 747 |
| N=32 | MoCA | 47 | 36 | 79 | 1894 |
| | PartCrafter | 48 | / | 142 | 2018 |

We conduct a runtime analysis of our method in comparison with PartCrafter (Lin et al., 2025), the most closely related baseline. Specifically, we report the inference-time breakdown of local attention, the routing procedure (MoCA only), and global attention, as well as the total runtime of a single forward pass. All experiments are performed on a single H20 GPU, with each part represented by 1024 tokens. The results are summarized in Table 4.

Under a large number of parts (N = 32), the global attention calculation of our model is significantly faster than that of PartCrafter (approximately half the runtime). However, because our routing procedure involves extensive sorting and indexing operations implemented purely in PyTorch, it currently introduces a substantial runtime overhead. We plan to further optimize this component to improve overall performance.

### A.4 ADDITIONAL QUALITATIVE RESULTS

In Figure 10, we provide a bunch of qualitative results of our model on both part-level object generation and instance-level complex scene generation.

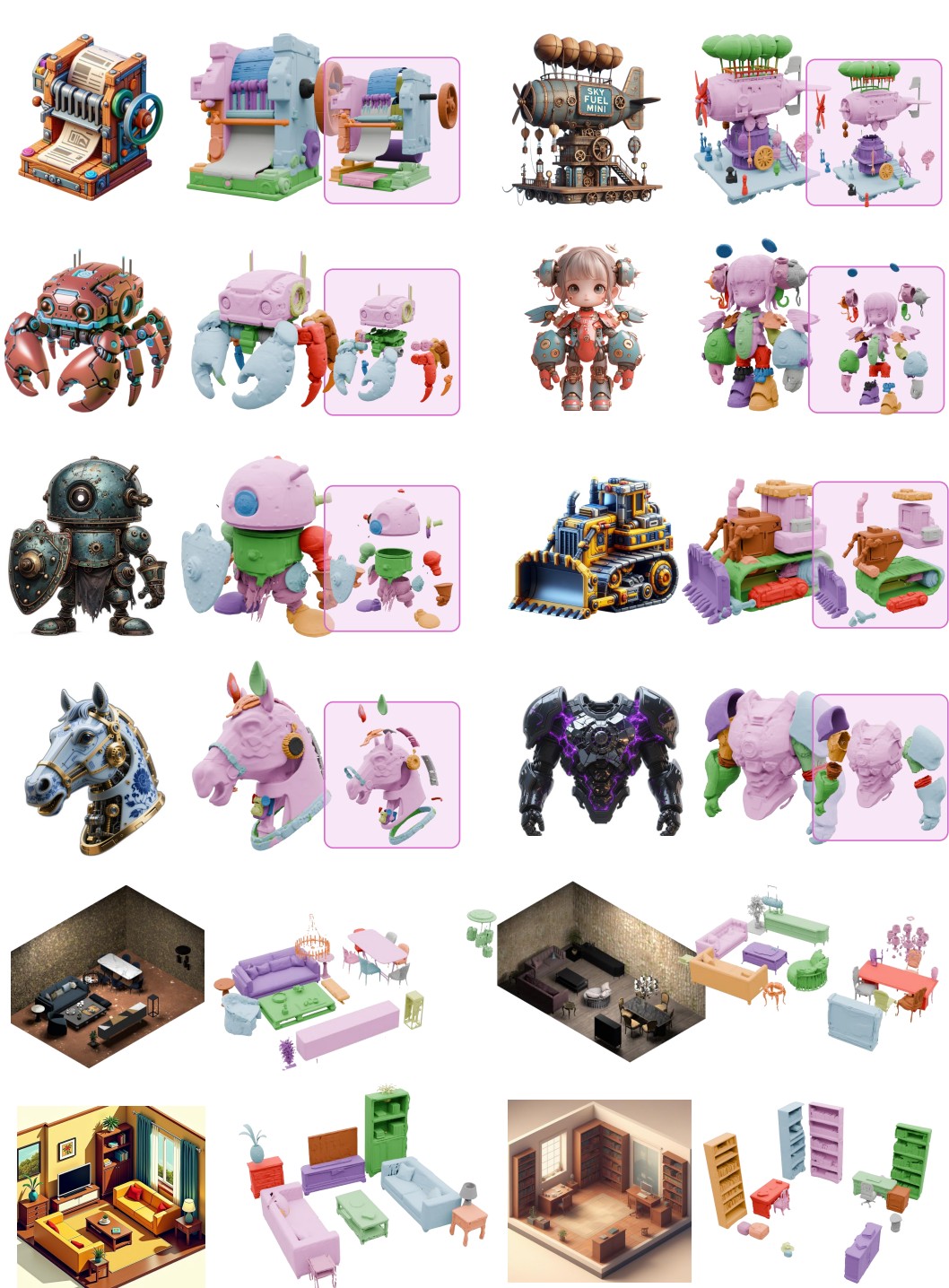

Figure 10: **Additional qualitative results of our method**.

