# OpenReview forum: "MoCA: Mixture-of-Components Attention for Scalable Compositional 3D Generation"
_ICLR.cc/2026/Conference — Submitted to ICLR 2026_

### Official Review · Reviewer_ELxf · 2025-10-28

**Soundness:** 3
**Presentation:** 3
**Contribution:** 3
**Rating:** 6
**Confidence:** 4

**Summary:**

This paper presents a new method for compositional 3D generation. The authors identity that part-aware 3D generation suffers from poor scalability due to the quadratic complexity of attention-based models. The authors therefore proposed to use mixture-of-experts to route the most important components and compress distant components to achieve linear complexity. The authors evaluated their method on both object-level and scene-level 3D generation tasks to demonstrate the effectiveness of their methods.

**Strengths:**

- The paper is well-motivated and technically sound.
- The use of MoE in 3D generation is interesting and effective.

**Weaknesses:**

- The generated results in Figure 3 are less detailed compared to baselines like PartCrafter. The surface is much smoother.
- The comparison is relatively limited compared to papers like PartCrafter, which compared on different datasets like Objaverse, ABO etc.
- Visual results and 3D results are fairly limited. Would love to see more 3D rendering results.

**Questions:**

I would love to see the authors provide more comparison and video results. I would also like to see the explanations of loss of details in reconstruction.

---

> ### Author Response · Authors · 2025-11-20
> **Response to Reviewer ELxf**
>
> Thank you for the constructive and valuable review. We are glad that you found the method well-motivated and technically sound. Below we address your specific comments and hope our response helps address your concerns.
>
> ### **Q1: Less detailed generation compared to baseline**
> **A1**: It is caused by different data preprocessing between ours work and PartCrafter. As a premise, we should know that the base 3D object generator is trained on watertight meshes. PartCrafter is trained on part surfaces that are directly sampled on raw meshes (without ensurance of watertightness). It preserves the sharp edges of surfaces, however, broken surfaces could be generated due to the involvement of non-watertight data. Inversely, we convert all training data into watertight for shape soundness. But due to the large amounts of data, the watertight conversion is conducted under limited spatial resolution, leading to minor loss of shape details.
>
> ### **Q2: Concerns about limited benchmark for comparison**
> **A2**: In our work, we mainly utilize PartObjaverse dataset for object task evaluation, which is a strong benchmark composed of 200 complex objects across diverse categories from Objaverse. Here, we supplement the evaluation results of our method and baseline methods on ABO dataset, please refer to Table 1 in our revision. Our model still achieves superior scores on all metrics.
>
> ### **Q3: Suggestion for supplementing more visual results**
> **A3**: We have uploaded the rendered videos of several complex test cases that are unseen in our main paper, please download the supplementary material to check it. More visual results will be hosted on our project page in the final version.

---

### Official Review · Reviewer_oLgj · 2025-10-31

**Soundness:** 2
**Presentation:** 1
**Contribution:** 1
**Rating:** 2
**Confidence:** 4

**Summary:**

This paper investigates part-composed 3D object and scene generation, an important and timely problem in 3D vision. The approach leverages the availability of high-quality modern 3D datasets. The authors propose a transformer-based model (arguably over-complicated) trained with flow matching to learn the distribution of shape latents. Each part is encoded and decoded in SDF representation, while image and text conditions can be incorporated into the diffusion transformer. Only a few examples are shown, also demonstrating synthetic image to compositional 3D generation.

**Strengths:**

- Although the method itself is standard, the results appear strong, largely due to the high quality of current datasets rather than novel modeling contributions.
- The focus on structured, compositional 3D generation is a meaningful and valuable research direction that deserves further attention.

**Weaknesses:**

- The transformer design lacks novelty and is quite widely used and studied in the literature of scene generation and part-based 3D object generation. Given the small latent space (fewer than 100 parts), any sufficiently large transformer could model the distribution; the architectural choices do not seem critical. I would believe this is a "fake" contribution.
- The reported results are not diverse, raising concerns about overfitting to the dataset.
- The paper only evaluates on synthetic data and generated images no real image experiments are provided.

**Questions:**

- Main question: What is the real contribution of this paper to the community?
- Secondary: How does the method ensure diversity and handle real data?

---

> ### Author Response · Authors · 2025-11-20
> **Response to Reviewer oLgj**
>
> We appreciate the reviewer’s time and the opportunity to clarify the contributions and technical novelty of our work. We respond to each point below and hope our response helps address your concerns.
>
> ### **Q1: Concerns about novelty and real contribution of this paper**
> **A1**: The core of our method design is a novel attention mechanism based on *component-level sparse routing.* We make the model learn inter-component dependencies and compress the distant components to reduce computational budget.
>
> To our best knowledge, we are *the first* to do this in the literature of compositional 3D generation. Since the reviewer claims that this design “is quite widely used and studied in the literature of scene generation and part-based 3D object generation”, could the reviewer please provide several references that we neglected before?
>
> While a single component requires at least 512 tokens to represent, 100 components (as mentioned by the reviewer) would lead to a large number of tokens (50K). The context of such a length requires tremendous resources to conduct training and inference. We argue that the reviewer should not overlook the value of the research works on efficient computation.
>
> ### **Q2: Concerns about diversity and potential overfitting**
> **A2**: Our task is image-to-3D compositional reconstruction, where the goal is faithfulness to the input, not unconditional diversity. Thus, diversity metrics are not directly applicable. As shown in Table 1 and Table 2, our method achieves the best scores on the metrics measuring geometry fidelity.
>
> Regarding concerns about overfitting to training data, we highlight that:
>
> - Our model generalizes well to AI-generated images, which are not part of the training distribution.
> - These results exhibit consistent geometry and stable decomposition, suggesting that the model does not simply memorize training samples.
>
> ### **Q3: Requirement of handling real data**
> **A3**: We agree that evaluation on real images is meaningful. In the revision, we have added results on real-world photographs (Fig. 5), demonstrating that our method can generalize beyond synthetic datasets.
>
> It is also worth noting that existing baselines in compositional 3D generation rarely benchmark on real images, due to the domain gap between internet images and structured part-based 3D datasets. Thus, the absence of such evaluation is a common limitation rather than a major weakness of our approach.
>
> ---
> We hope these clarifications help address your concerns. We sincerely appreciate your detailed comments and would be happy to continue the discussion.

---

### Official Review · Reviewer_Dfrs · 2025-10-31

**Soundness:** 4
**Presentation:** 4
**Contribution:** 3
**Rating:** 8
**Confidence:** 4

**Summary:**

This paper introduces MoCA, a compositional 3D generative model for efficient, scalable, and accurate compositional modeling of 3D objects and scenes.

**Strengths:**

1. The proposed Mixture-of-Components Attention is well motivated for addressing the quadratic global attention cost.

2. Complete ablations: All design choices (compression, gating, activation, multi-head routing) are completely ablated.

3. Strong performance: better experimental results have been observed against baselines like PartPacker, PartCrafter, MIDI

**Weaknesses:**

1. No apperance: It seems that all methods, including baselines, only generate meshes without textures, which might limit the real-world applications. Can the authors provide more details about this?

**Questions:**

Runtime analysis: It would be better to include a breakdown of runtime about different procedures of the proposed pipeline and compare against other baselines to demonstrate the efficiency.

---

> ### Author Response · Authors · 2025-11-20
> **Response to Reviewer Dfrs**
>
> Thank you for the positive and encouraging feedback, and for highlighting the strengths of our work. We address your comments in detail below and hope our response helps address your concerns.
>
> ### **Q1: Concerns about appearance generation**
> **A1**: Yes, our model along with the baseline methods are all focused on shape decomposition, since the base 3D object generator is also shape-only. How to generate consistent texture for all parts of a single object remains an open question, and we will delve to it in our future research.
>
> ### **Q2: Suggestion for adding runtime analysis**
> **A2**: We have included the runtime analysis of our model against PartCrafter (the most related baseline method). Please refer to Table 4 in the appendix of our revision. Specifically, we report the inference-time breakdown of local attention, the routing procedure (MoCA only), and global attention, as well as the total runtime of a single forward pass. All experiments are performed on a single H20 GPU, with each part represented by 1024 tokens.
>
> Under a large number of parts (N = 32), the global attention calculation of our model is significantly faster than that of PartCrafter (approximately half the runtime). However, because our routing procedure involves extensive sorting and indexing operations implemented purely in PyTorch, it currently introduces a substantial runtime overhead. We plan to further optimize this component to improve overall performance.

---

> > ### Comment · Reviewer_Dfrs · 2025-11-26
> > **Regarding appearance generation**
> >
> > Thanks for your rebuttal.
> >
> > Regarding the apperance generation, I did see two papers:
> > 1. PartGen: Part-level 3D Generation and Reconstruction with Multi-View Diffusion Models (https://silent-chen.github.io/PartGen/)
> > 2. MIDI: Multi-Instance Diffusion for Single Image to 3D Scene Generation (https://github.com/VAST-AI-Research/MIDI-3D)
> >
> > The authors compared with MIDI but without the texture generation because the base 3D object generator is shape-only.
> >
> >  I am wondering if the proposed method can be easily exteneded to other models like Trellis (https://arxiv.org/abs/2412.01506). Some preliminary explanations would be acceptable.

---

> > > ### Author Response · Authors · 2025-11-26
> > > **Response to Reviewer Dfrs**
> > >
> > > **Regarding appearance generation**
> > >
> > > Thank you very much for the helpful references and for pointing this out. We clarify our perspective as follows:
> > >
> > > *PartGen* is a multi-stage pipeline. In the last step of its pipeline, it lifts the multi-view images of completed parts to textured 3D using a pretrained reconstruction model, which requires simultaneous modeling of both shape and appearance. Yet their code is not released.
> > >
> > > *MIDI* shows an application to generate textured 3D objects in the scene by stitching another multi-view image generator *MV-Adapter* (https://arxiv.org/abs/2412.03632) after its main pipeline. For fine-grained part-level generation, however, multi-view image generators are typically trained on monolithic object images and do not naturally align with part-decomposed inputs, making the extension less straightforward.
> > >
> > > In our view, the most principled path to part-level textured generation is to upgrade the base 3D generator to jointly model shape and appearance (e.g., through a unified SDF + radiance/texture latent space), and then apply MoCA (or PartCrafter/MIDI-style pipelines) on top. This would yield more consistent and scalable textured-part generation than multi-stage methods that rely on external lifting modules or multi-view generators.
> > >
> > > We agree this is an exciting future direction, and your suggestion is very helpful.
> > >
> > > **Regarding extending our proposed method to TRELLIS**
> > >
> > > Thank you for raising this interesting question. We believe MoCA can be extended to Trellis with relatively light modifications.
> > >
> > > The core ideas of MoCA are:
> > >
> > > 1. Explicitly modeling inter-component dependencies, and
> > > 2. Allowing each component to selectively attend to important components with full features, while using compressed features for the others.
> > >
> > > These principles are architecture-agnostic and do not depend on the VecSet representation itself.
> > >
> > > Trellis differs from VecSet-based models mainly in its structured latent space (sparse voxel grids) and its two-stage pipeline. The main adaptation would be in **how we compress components**:
> > >
> > > - In VecSet, latent tokens are unordered, so compression is done through learnable queries in cross-attention.
> > > - In Trellis, the latent representation is spatially structured, enabling direct spatial pooling or convolutional downsampling as a more natural compression mechanism.
> > >
> > > Once the compression operator is adapted, the Mixture-of-Components Attention can be inserted into the DiT modules of Trellis’s stages without major changes, since the routing mechanism (importance scoring, top-K selection, multi-head routing) operates independently of the specific latent parameterization.
> > >
> > > Therefore, we believe MoCA can be incorporated into Trellis’s architecture with minimal conceptual changes and only modest engineering effort.

---

### Official Review · Reviewer_sfEm · 2025-10-31

**Soundness:** 3
**Presentation:** 3
**Contribution:** 2
**Rating:** 4
**Confidence:** 3

**Summary:**

TLDR: using routers to select top relevant tokens between parts to avoid full attention, enables larger number of parts generation.

This paper proposes a new approach to the image to 3D parts problem. It tries to generate more parts than previous work by using routers to select top relevant tokens between parts to avoid full attention, which is inspired by the router mechanism in the Mixture-of-Experts (MoE) methods.

This can effectively reduce the memory requirement and hence support up to 30 components generation.

**Strengths:**

- Good motivation: The most challenging aspect of part generation is having a large number of parts. This works tries to improve performance on such important task.
- Good results. The method can generate more parts than previous work.
- Sound technical approach. The general design of the method is sound. And the idea of using routers is interesting.

**Weaknesses:**

- Complicated system. The system seems to be very complicated, and the technical details are hard to read. Method pipeline figure is challenging to understand - Maybe a better way is to decompose the figure into multiple figures so it is easier to understand part by part.
- Limited insight. Although the Routing mechanism seems valid, it is a general method and the author does not further utilize properties that are unique to 3D part structures.

**Questions:**

- Why the part geometry quality degrade significantly when the number of parts are increased? In figure 6, shapes start to be scattered and broken for part number to be 30.
- Also in figure 6, we show the parts clearly get disconnected from each other (the feet and the legs), please provide more analysis on this limitation and give potential ideas to solve
- What if you don't use routing scheme at all? How do you compare your model removed the routing mechanism. The results should be better since it will use full attention, but what will be the difference, and how do you compare the results when you have same number of parts, like 30, but with smaller number of tokens per part so the parts can fit in the memory, will it have similar results as the scattered ones you have shown in figure 6?

If the author can better understand how routing contributes to the existing approach, I would consider to raise my score.

---

> ### Author Response · Authors · 2025-11-20
> **Response to Reviewer sfEm**
>
> Thank you very much for the thoughtful and constructive review. We are glad that you found our motivation and technical design sound. We address your questions point-by-point below and hope our response helps address your concerns.
>
> ### **Q1: Suggestion for better pipeline illustration**
> **A1**: We thank the reviewer for the suggestion. We have decomposed the pipeline diagram into two new figures (overview of the pipeline, details of Mixture-of-Components Attention) for easier understanding, please refer to Fig. 2 and Fig. 3 in our revision.
>
> ### **Q2: Limited insight of routing mechanism**
> **A2**: Given that a model relying on full attention over all part tokens cannot scale to a large number of parts, our goal is to design an attention mechanism that selectively allocates more computation to salient context while reducing cost on less-relevant context, which is the fundamental principle underlying sparse attention methods.
>
> For 3D part data, structural and semantic dependencies naturally arise among parts (e.g., a human hand is strongly related to the arm but only weakly related to the head). Motivated by this unique property of 3D part data, we design our routing module to explicitly learn and exploit such inter-part dependencies.
>
> Although our routing module draws inspiration from the general design of MoE routers, it is theoretically distinct: our router performs **part-level routing**, sending each part to its top-K most relevant parts, whereas MoE routers perform **token-level routing**, sending each token to its top-K selected experts. Therefore, our method is not a simple application of standard MoE routing; rather, it introduces a new architectural innovation tailored to the compositional 3D generation setting.
>
> ### **Q3: Concerns about degraded quality when the number of parts are increased**
> **A3**: The main reasons for the degraded geometry quality under large part number are as follows:
>
> - As mentioned in paragraph “Limitations and Future Works” of our paper, when the object (normalized in unit cube) is decomposed to large number of parts, the volume of each part is quite small. Decoding tiny shapes would introduce additional error due to large deviation from the VAE’s training phase.
> - Modeling complex composition is naturally much more difficult for the model.
> - Although the efforts in our data curation, the data with large number of parts are still noisier than those simple data. Moreover, data with large number of parts is of less proportion in the whole dataset. These data-level issues make it more challenging for the model to learn complex composition.
>
> Our work mainly contributes to extend the context of parts by a large ratio (2x) through a novel efficient attention mechanism. We will further polish our model’s performance on large part numbers with more engineering efforts on the above mentioned issues.
>
> ### **Q4: Concerns about disconnected parts**
> **A4**: Same as the baseline method, PartCrafter, our model generates 3D assets as plausible composition of semantically meaningful components from a single image. The feet and the legs are both semantically meaningful parts, and the model generates them separately is reasonable. In figure 6, it also shows cases where feet and legs are treated as a whole, which reveals the diversity of our model’s generation.
>
> To add more controllability to the generated parts, we can finetune the pretrained model with injection of additional control signals (e.g., part mask, bounding box).
>
> ### **Q5: Ablation on removing routing mechanism**
> **A5**: We have appended the ablation results of removing our router module and conducting uniform global attention on only compressed part tokens, please refer to Table 3O and Figure 8 of our revision for the quantitative and qualitative results. The results show a significant performance drop without the explicit modeling of inter-part dependencies by our router and the attendance of important parts' full features.

---

> > ### Comment · Reviewer_sfEm · 2025-11-27
> >
> > Thanks for the detailed response, my concerns are addressed.

---

> > > ### Author Response · Authors · 2025-11-27
> > >
> > > Thank you very much for the follow-up and for letting us know that your concerns have been addressed.
> > > We are truly grateful for your thoughtful feedback throughout the process.
> > > If there is any remaining point where further clarification could be helpful, we would be very happy to elaborate.
> > >
> > > Thanks again for your time and constructive engagement in the discussion.

---

### Author Response · Authors · 2025-11-20
**Response to all reviewers**

We sincerely thank all reviewers for their thoughtful feedback and constructive suggestions. We are encouraged that the reviewers find our motivation strong, the method technically sound, and the results promising—especially the ability to significantly scale compositional 3D generation.

Following the reviewers’ comments, we have revised the paper extensively. All changes are highlighted in **blue** in the updated manuscript. The main improvements are summarized below:

### **Key Revisions**

- **Clearer methodology presentation:**

    We reorganized the pipeline illustration into two new figures (an overall pipeline view and a detailed depiction of Mixture-of-Components Attention ) to improve clarity and readability. *(Fig. 2–3)*

- **Expanded ablations and analysis:**

    Additional ablation studies are included to analyze the effect of removing the routing module and performing uniform attention on compressed tokens. These results strengthen the necessity of explicitly modeling inter-part dependencies. *(Table 3O, Fig. 8)*

- **Runtime analysis:**

    We added a detailed breakdown runtime comparison with the most relevant baseline, PartCrafter. *(Table 4 in Appendix)*

- **Real image experiments:**

    We included new experiments on real-world images to demonstrate the generalizability of MoCA beyond synthetic data. *(Fig. 5)*

- **Additional benchmarks:**

    We added supplementary evaluations on ABO to provide broader comparison. *(Table 1)*

- **Improved qualitative visualizations:**

    We uploaded rendered multi-view videos for several complex test cases apart from those cases shown in manuscript.


We appreciate all reviewers’ input, which helped us greatly improve the paper. Below we address each comment in detail in a point-by-point manner. We welcome further discussion.

---

> ### Author Response · Authors · 2025-11-27
> **Looking forward to your further comments**
>
> Dear Reviewers,
>
> Thank you again for the time and effort you have devoted to reviewing our submission. We have carefully addressed the questions and concerns raised across the reviews, and the updated manuscript and supplementary material now incorporate all corresponding revisions.
>
> We sincerely appreciate your thoughtful feedback and would be grateful for any further comments or assessments you may have as we move forward in the discussion phase.
>
> Best regards,
>
> The Authors

---

### Author Response · Authors · 2025-12-01
**Brief summary for the Area Chair**

Dear AC,

Thank you for taking over the assessment in light of the recent AC re-assignment. We provide this concise comment to assist your evaluation.

We first thank all reviewers for their time and constructive comments, especially their agrees that:

- The scalability challenge in compositional 3D generation is important.
- The proposed Mixture-of-Components Attention is well-motivated (sfEM & Dfrs & ELxf), technically sound (sfEm & ELxf) and interesting (sfEm & ELxf).
- Our work demonstrates strong performance against baseline methods (sfEm & Dfrs) and conducts complete ablations (Dfrs).

In the following, we would like to summarize the status with respect to each reviewer before the incident:

- Reviewer sfEm (original rating 4): After rebuttal, explicitly stated “*my concerns are addressed*.”
- Reviewer Dfrs (original rating 8): Continued technical discussion regarding extensions (appearance, extending to TRELLIS).
- Reviewer ELxf (original rating 6): No unresolved concerns after our clarification on the question and supplementary of requested comparisons and visual results.
- Reviewer oLgj (original rating 2): Raised substantive concerns about novelty, diversity, and real-data generalization. In response, we provided detailed point-by-point clarifications and added experiments on real photographs. Regrettably, the discussion was halted before the reviewer could respond to our rebuttal.

We fully understand that, due to the recent platform issue, ACs now face a significantly increased workload. We sincerely appreciate your effort under these unusual circumstances and would be grateful if you could, despite the added load, consider our strengthened results and resolved reviewer discussions in forming your final assessment.

Thank you very much for your time and attention.

Best regards,

The Authors

---

### Meta-Review · Area_Chair_fhFY · 2025-12-30

**Summary:**

The reviewers pointed out that the paper mainly suffers from the following issues:
1.	Overly complex technical details and a method pipeline figure that is difficult to interpret.
2.	A lack of evaluation and comparison of model efficiency.
3.	Insufficiently convincing evidence of novelty and effectiveness, requiring more diverse benchmark results and comparisons on real data.

**Reviewer Concerns:**

While most concerns have been addressed by the authors, the paper lacks convincing explanations and ablation studies regarding the design of the method’s architecture and the use of the routing mechanism. In addition, evaluations on a more diverse set of real-world images are expected to demonstrate the method’s effectiveness.

**Reviewer Scores:**

Most reviewers tend to maintain their initial scores.

---

### Decision · Program_Chairs · 2026-01-26

Reject